# Competition among variants is predictable and contributes to the antigenic variation dynamics of African trypanosomes

**Douglas O. Escrivani©, Viktor Scheidt¤ª©, Michele Tinti, Joana Faria¤ᵇ, David Horn ©\***

Wellcome Centre for Anti-Infectives Research, School of Life Sciences, University of Dundee, Dow Street, Dundee, United Kingdom

© These authors contributed equally to this work.
¤a Current address: Dr. Brill & KEBOS GmbH, Institute for Hygiene and Microbiology, Hamburg, Germany.
¤b Current address: York Biomedical Research Institute, Department of Biology, University of York, Heslington, York, United Kingdom.
\* d.horn@dundee.ac.uk

**Data Availability Statement:** The high-throughput RNA-seq data generated in this study have been deposited in the Short Read Archive (SRA) under

## Abstract

Several persistent pathogens employ antigenic variation to continually evade mammalian host adaptive immune responses. African trypanosomes use variant surface glycoproteins (VSGs) for this purpose, transcribing one telomeric *VSG* expression-site at a time, and exploiting a reservoir of (sub)telomeric *VSG* templates to switch the active *VSG*. It has been known for over fifty years that new VSGs emerge in a predictable order in *Trypanosoma brucei*, and differential activation frequencies are now known to contribute to the hierarchy. Switching of approximately 0.01% of dividing cells to many new VSGs, in the absence of post-switching competition, suggests that VSGs are deployed in a highly profligate manner, however. Here, we report that switched trypanosomes do indeed compete, in a highly predictable manner that is dependent upon the activated *VSG*. We induced *VSG* gene recombination and switching in *in vitro* culture using CRISPR-Cas9 nuclease to target the active *VSG*. *VSG* dynamics, that were independent of host immune selection, were subsequently assessed using RNA-seq. Although trypanosomes activated *VSGs* from repressed expression-sites at relatively higher frequencies, the population of cells that activated minichromosomal *VSGs* subsequently displayed a competitive advantage and came to dominate. Furthermore, the advantage appeared to be more pronounced for longer VSGs. Differential growth of switched clones was also associated with wider differences, affecting transcripts involved in nucleolar function, translation, and energy metabolism. We conclude that antigenic variants compete, and that the population of cells that activates minichromosome derived *VSGs* displays a competitive advantage. Thus, competition among variants impacts antigenic variation dynamics in African trypanosomes and likely prolongs immune evasion with a limited set of antigens.

accession codes PRJEB40415 [https://www.ncbi.nlm.nih.gov/bioproject/PRJEB40415] for the main time-course, and PRJNA852534 [https://www.ncbi.nlm.nih.gov/bioproject/PRJNA852534] for the independent 5- and 25-day samples and switched clones. Scripts to align the RNA-seq data and to compute read-counts have been deposited in GitHub [https://github.com/mtinti/VSG_Competition/] and Zenodo [https://doi.org/10.5281/zenodo.7942978].

**Funding:** The work was funded by Wellcome Investigator Awards to D.H. [100320/Z/12/Z and 217105/Z/19/Z] and a Wellcome Ph.D. studentship award to V.S. [203816/Z/16/Z]. The funders had no role in study design, data collection and analysis, decision to publish, or preparation of the manuscript.

**Competing interests:** The authors have declared that no competing interests exist.

## Author summary

Human and animal parasites, including those that cause sleeping sickness, malaria and giardiasis, have evolved effective mechanisms to counter host adaptive immunity. These parasite genomes typically incorporate large gene-families encoding variant surface antigens that are deployed during infection, allowing parasites to remain 'one step ahead' of host immune responses. It remains unclear, however, how these reservoirs of surface antigens are effectively deployed. We generated a complex mixture of African trypanosome variants and quantitatively monitored their dynamic behavior over time. The results confirm differential activation rates, and also reveal differential growth rates, which are predictable. Differential growth can facilitate persistence through sparing, rather than profligate, presentation of variant antigens.

## Introduction

The African trypanosome, *Trypanosoma brucei*, is a protozoan parasite, transmitted by tsetse flies, that causes persistent diseases in both humans and livestock; sleeping sickness and nagana, respectively. Exclusively extracellular bloodstream and tissue-resident trypanosomes can divide every 6–7 h, although substantially slower growth has been reported for adipose tissue forms [1]. These cells express a super-abundant variant surface glycoprotein (VSG), that forms a dense coat on each cell; comprising approximately 10% of total cell protein [2]. Indeed, translation in each *T. brucei* cell produces up to 80,000 VSG molecules every minute [3]. In the advanced second stage of infection, trypanosomes breach the blood-brain barrier and invade the central nervous system, causing severe and, without treatment, often lethal disease. These trypanosomes undergo antigenic variation by spontaneously switching the VSG, exploiting a reservoir of hundreds of intact *VSG* genes, even assembling mosaic *VSGs* from incomplete *VSGs* [4,5], and thereby avoiding clearance by the host's adaptive immune system. The consequent and ongoing 'arms race' at the parasite-host interface, involving antigenically distinct VSGs, and anti-VSG antibodies, is the result of two of the most prolific protein diversifying mechanisms in biology [6].

Only one VSG is expressed at a time by each *T. brucei* cell [7]. The active *VSG* is transcribed in one of fourteen alternative sub-telomeric 'bloodstream' expression sites (ESs) in the '427' strain [8] and switching is typically triggered by spontaneous DNA double-strand breaks that result in homologous recombination and gene conversion, replacing the active VSG [9–12]. In the first weeks of infection, donor *VSG* templates are typically from repressed ESs or from the ends of approximately 100 minichromosomes [13–15]. Repressed ESs, since they have promoters, can also be activated *in situ* [7]. Mosaic VSGs typically arise at low frequency and appear later during infection [4,5]. Thus, *T. brucei* is a remarkably persistent parasite that, as a result of antigenic variation, sustains host immune evasion for years or even decades [16].

Based on published switching rates that typically range from $10^{-3}$ to $10^{-4}$ per cell division [17], African trypanosomes could simultaneously present large numbers of VSGs to the host immune system in a highly profligate manner, which could rapidly exhaust the repertoire. Even a low parasitemia of one million cells, switching at a rate of $10^{-4}$ per cell division, has the potential to simultaneously present up to 400 different VSGs every day. Indeed, approximately 80 [5] to 190 VSGs [18] have been detected simultaneously *in vivo* in mouse models. The long-term persistence of African trypanosome infections, however, indicates that more complex mechanisms are likely at play. Variants may grow at different rates and may fail to elicit an

effective immune response until they constitute a substantial proportion of the overall parasitemia, for example [19].

It first became clear in the 1960s that *T. brucei* VSGs appear in a predictable order *in vivo* [20–22]. Further illustrating this point, several populations, all isolated on day eighteen of a chronic infection, were found to have independently activated the same VSG [23,24]. It has subsequently become clear that parasite-intrinsic factors contribute to the frequency of *VSG* activation. In particular, *VSG* location is important; a telomeric variant sharing more extensive flanking homology with the active variant, including upstream tracts of '70-bp repeats' [13], is more likely to be activated by recombination, for example [25–27]. Beyond differential *VSG* activation frequency, roles for switch intermediates [28] and density dependent parasite differentiation [29] have also been considered in impacting antigenic variation dynamics. A more recent mathematical model [19], and analysis of data from mouse infections [5], suggested that *T. brucei* parasites expressing different-length VSGs grow at different rates [19]. Indeed, evidence supporting the view that variants compete was initially reported more than forty years ago [22,30].

Here, we use CRISPR-Cas9 to trigger VSG switching and to generate a heterogeneous mixture of antigenic variants in cell culture. We then quantitatively monitor VSG abundance and dynamics over time using RNA-seq. Our findings reveal antigenic variation dynamics that are driven by the preferential activation of expression site derived *VSGs* and a subsequent competitive advantage of populations of cells expressing minichromosome derived *VSGs*. Further analysis of switched clones revealed VSG length differences and a transcriptome pattern that coincided with differential growth. Our findings explain features of *VSG* population dynamics that have been observed in mouse models and that likely allow these parasites to prolong immune evasion using a limited set of variant antigen genes.

## Results

### Cas9-induced DNA breaks trigger antigenic variation

Each *T. brucei* VSG displays a number of characteristic features (Fig 1A). An *N*-terminal signal sequence targets the protein to the secretory pathway and a *C*-terminal glycosylphosphatidylinositol (GPI) signal directs post-translational membrane anchoring. Despite a remarkable level of primary sequence divergence, a pattern of cysteine residues [31] directs protein folding and maintains a characteristic structure [32]. A number of different *N*- (A-C) and *C*-terminal domain type (1–4) VSGs are derived based on these features and two examples are shown in Fig 1A; *C*-terminal domain types 1 and 3 typically contain eight Cys residues and are longer than types 2 and 4, which typically contain four Cys residues [31].

To assess VSG population dynamics, we sought to establish a complex antigenically variant *T. brucei* population in *in vitro* cell culture. Prior studies employed the I-*Sce*I meganuclease to introduce a DNA break adjacent to the active *VSG* and to initiate efficient VSG switching [9, 10, 12, 13]. These studies also indicated that resection from the break promoted recombination within upstream tandem 70-bp repeats, followed by VSG gene conversion (see Fig 1B). We more recently developed an inducible CRISPR-Cas9 system in *T. brucei* [33], and reasoned that this would likely provide a more versatile approach for programming DNA breaks that trigger VSG switching, without prior modification of the target sequence. We first selected three distinct Cas9-single guide RNA (sgRNA) target sequences within the active bloodstream *VSG* expression site (BES1) in our experimental strain; two target sites located between the 70-bp repeats and the active *VSG* and one within the active *VSG* coding sequence (Fig 1B). *VSG* replacement is expected to remove these Cas9-sgRNA target sites, thereby preventing further *VSG* switching (Fig 1B).

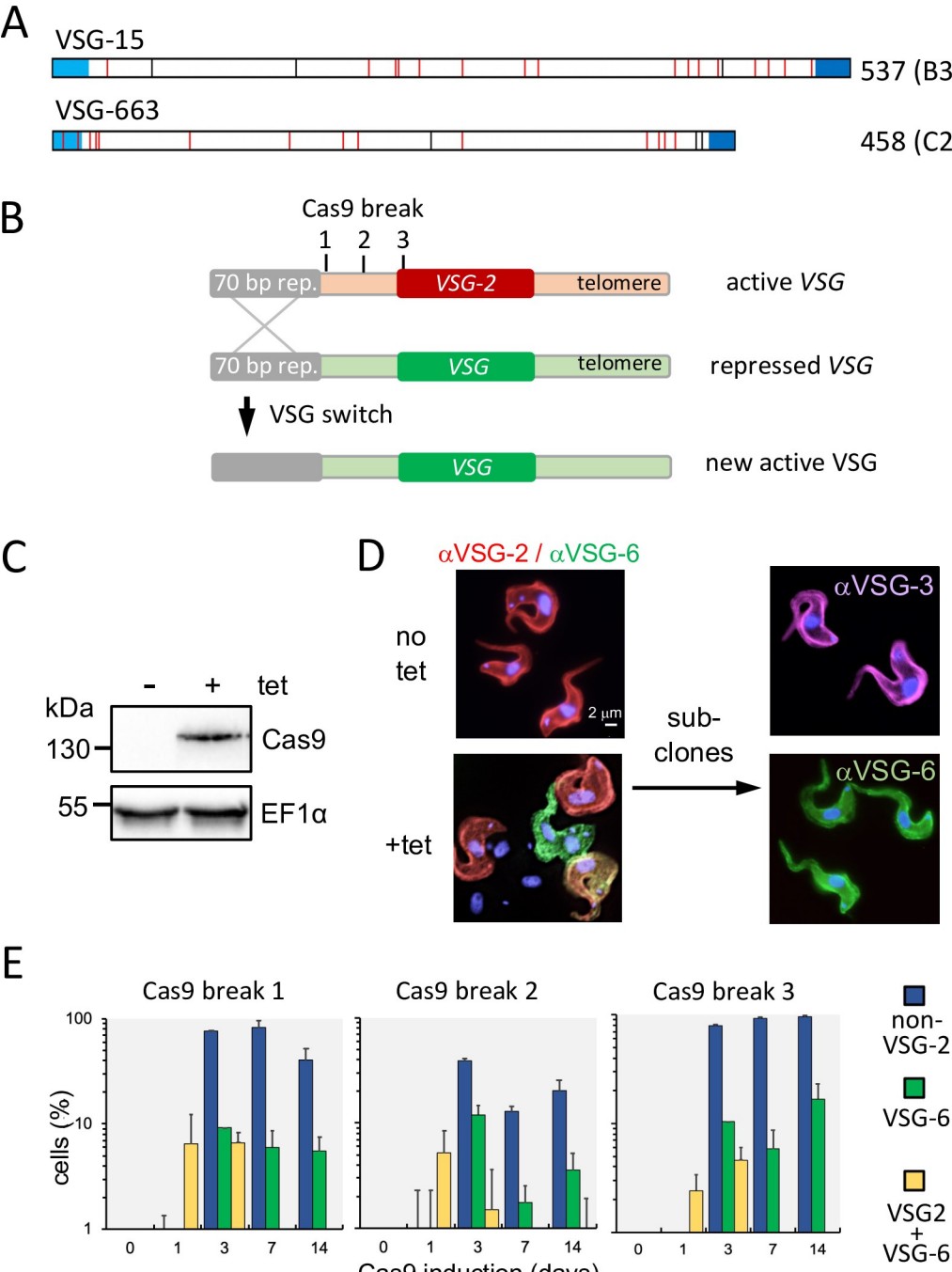

**Fig 1. Cas9-induced DNA breaks trigger antigenic variation.** (A) The schematic shows two examples of VSGs with salient features highlighted; the *N*-terminal signal sequence, Cys residues (red), predicted N-glycosylation sites (black) and *C*-terminal GPI anchor signal. Length, in amino acids, and class (B3 or C2) are indicated. See S2 Fig for more examples. (B) The schematic shows the active telomeric *VSG* (*VSG-2*) and the sites targeted by Cas9-sgRNAs to introduce DNA double-strand breaks. Recombination within 70-bp repeats triggers gene conversion and duplicative replacement of *VSG-2* with a new *VSG*. (C) The protein blot shows robust inducible expression of Cas9. EF1α was used as loading control. (D) Immunofluorescence microscopy reveals switched and intermediate switching cells following Cas9-induction. Two switched sub-clones assessed by immunofluorescence microscopy are also shown. (E) An immunofluorescence microscopy time-course assay, with continuous Cas9 induction. Two biological replicates for each sgRNA. n > 90 cells for each sample at each time-point.

Protein blotting confirmed that Cas9 was tetracycline-inducible (Fig 1C). Using one sgRNA strain initially, we carried out immunofluorescence analysis to demonstrate a homogeneous population expressing VSG-2 prior to Cas9 induction, and a mixed population following Cas9 induction; cells that had switched to VSG-6 or another VSG, as well as switch intermediates, were detected (Fig 1D). By sub-cloning these populations and assessing a panel of clones using *VSG*-generic RT-PCR and Sanger sequencing, we demonstrated switching to different *VSGs*. Immunofluorescence analysis revealed stable expression of new switched VSGs; two example clones are shown in Fig 1D, and no subsequent switching was detected in pairs of clones expressing VSG-3 or VSG-6 (n > 700 cells). To compare the efficiency of each sgRNA in triggering antigenic variation, we induced Cas9 and monitored VSG-2 and VSG-6 expression over time (Fig 1E); duplicate clones were >99.5% homogeneous for VSG-2 expression prior to Cas9-induction. The analysis revealed that Cas9 induction triggered high-efficiency VSG switching when combined with each of the three sgRNAs tested; break-site 3, within the active VSG, generated the highest proportion of switched cells. We were able to detect switch intermediates with both VSG-2 and VSG-6 at the surface on day-1 and day-3 after Cas9-induction, but not before and not thereafter, while cells that no longer expressed detectable VSG-2 at the surface were first detected on day-3 (Fig 1E). Thus, our approach provided tightly regulated and highly efficient Cas9-based triggers for VSG switching.

## RNA-seq revealed relative switching frequency to distinct groups of *VSGs*

We selected sgRNA break-site 3, within the active VSG, for subsequent analyses (see Fig 1B and 1E). We induced Cas9 for three days and allowed cells to complete the switching process for a further two days. We then split these cells into three technical replicate cultures, took the first mRNA samples for RNA-seq (day-5) and then took subsequent mRNA samples for RNA-seq on days 9, 13, 17 and 25 (S1 Data). We also monitored VSG-2 and VSG-6 expression in these populations and, consistent with the analysis above (Fig 1E), found a high proportion of cells throughout the time-course that no longer expressed VSG-2 (Fig 2A). As expected, switch intermediates, with both VSG-2 and VSG-6 at the surface, were no longer detectable on day 5, nor thereafter; prior studies also indicated full replacement of VSG coats ~4.5 days after switching [34]. Thus, our RNA-seq analysis from day 5 to day 25 represented, as intended, a post-switching time-course.

RNA-seq reads were initially aligned at high-stringency [35] to the *VSG*-ESs [8] to determine whether switched cells typically replaced only the DNA segment containing the *VSG* at the active ES (see Fig 2B). As expected, and consistent with recombination within 70-bp repeats, multiple ES-*VSGs* are expressed in the day-5 populations, while the *ESAGs* in their respective ESs are not expressed (Fig 2C; BES3, 7 and 12 are shown). In contrast, the *ESAGs* in the initially active *VSG*-ES, BES1, are expressed (Fig 2C; top panel). Thus, transient Cas9-induction yielded cells that typically replaced the *VSG* at the active ES, while retaining the same set of transcribed *ESAGs* (see Fig 2B).

We next considered the cohort of expression site derived *VSGs* (ES-*VSGs*) and minichromosome derived *VSGs* (MC-*VSGs*) that could be monitored in our switched populations. The Lister 427 *T. brucei* strain used here is the most commonly used trypanosome model for studies on antigenic variation. This strain benefits from excellent annotation of a full set of sequenced *VSG*-ESs; in the form of intact cloned fragments of up to 60 kbp [8], and several annotated monocistronic *VSG*-ESs, typically expressed in the metacyclic stage [13, 36–40]; these closely related subtelomeric sequences typically remain unassembled in short-read genome sequencing projects [41]. Thus, we collected the set of ES-*VSG* sequences, together with a set of MC-*VSG* sequences, and RNA-seq reads were aligned at high-stringency to these

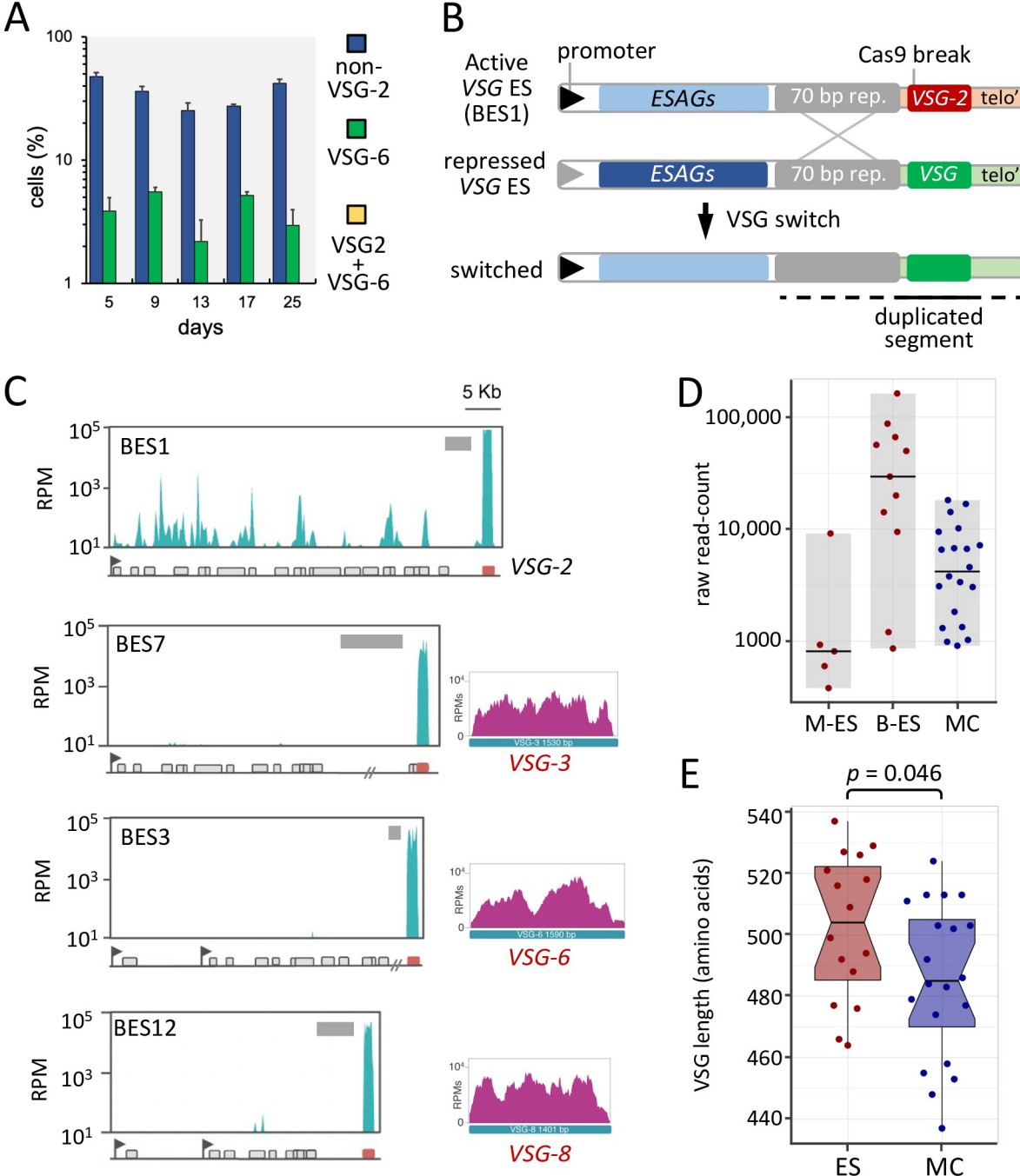

**Fig 2. RNA-seq confirms that switched cells retain common *ESAGs*.** (A) An immunofluorescence microscopy time-course assay, beginning on day-5; after three days of growth in tetracycline, to induce Cas9 with sgRNA-3, and a further two days after removal of tetracycline. Three technical replicates. n > 90 cells for each sample at each time-point. RNA was extracted from these samples for RNA-seq analysis. (B) The schematic shows the active telomeric *VSG*-ES (BES1) and the site targeted by sgRNA-3. Recombination with a silent bloodstream ES triggers replacement of *VSG-2* with a new ES-*VSG*, while retaining the BES1 *ESAGs*. (C) RNA-seq analysis of *VSG*-ESs. Only the *VSGs* (3, 6 and 8) are activated, but not their respective *ESAGs* (grey boxes below each read-mapping profile, in BES7, 3 and 12), as illustrated in B. Values expressed in RPMs (reads per million). Flags indicate *VSG*-ES promoters; grey blocks indicate the extent of the 70-bp repeats. (D) The graph shows *VSG* expression levels as determined by RNA-seq on day-5 as jittered dots; averages of three replicates. The number of VSGs is: metacyclic ES-*VSGs* (M-ES) = 5, bloodstream ES-*VSGs* (B-ES) = 11, minichromosomal *VSGs* (MC) = 20. The horizontal bars indicate the mean for each group. Values from S1 Data. (E) The graph shows the length of the activated *VSG* set as jittered dots. ES-VSGs, n = 16; MC-*VSGs* n = 20. The summary of the data is shown as a boxplot, with the box indicating the IQR, the whiskers showing the range of values that are within 1.5*IQR and a horizontal line indicating the median. The notches represent for each median the 95% confidence interval (approximated by 1.58*IQR/sqrt(n)). The *p* value was derived using a two-tailed *t*-test.

*VSGs*. Recognizing the challenge of working with a complex gene family, we inspected the read-mapping profiles for evidence of expression of the full coding sequence (CDS). *VSGs* for which mapped reads were restricted to the conserved *C*-terminal sequence (see S1 Fig) were removed. This yielded thirty-six activated *VSGs* (S2 Fig) that were readily detected by RNA-seq in our switched populations (see Materials and Methods); five metacyclic ES-*VSGs*, eleven polycistronic bloodstream ES-*VSGs* and twenty MC-*VSGs*. The read-mapping profiles for three example ES-*VSGs* are shown in Fig 2C (*VSGs* 3, 6 and 8 in BES7, 3 and 12, respectively).

Taking the *N*-terminal 400 amino acids predicted for each VSG, which comprises the divergent domain, and the *C*-terminal 100 amino acids, we constructed phylogenetic trees, which revealed a highly divergent set of *N*-terminal sequences (S2 Fig). Based on these analyses, and to improve mapping fidelity, we took only the first 1.2 kbp of each *VSG* CDS and remapped the RNA-seq data (S1 Data). An assessment of relative read-count in our earliest RNA-seq samples (day 5) then revealed differential activation frequencies for each *VSG* and for each cohort of *VSGs* (Fig 2D). Bloodstream ES-*VSGs* were on average eight times more likely than MC-*VSGs* to be activated and nineteen times more likely than metacyclic ES-*VSGs* to be activated. Prior to further analysis, we also assessed VSG length distribution, since a prior study [19] suggested a role for VSG length in driving antigenic variation dynamics in a mouse model [5]. The VSG length range seen in the set we detected by RNA-seq extends from 437 to 537 amino acids, a 23% differential (Fig 2E). Notably, ES-VSGs were significantly longer than MC-VSGs; seven of the eight longest VSGs (>515 aa) were from ESs, while the five shortest VSGs were from minichromosomes. We also note prior analysis here, indicating that all ESs appear to encode unique *VSGs* [8], and suggesting that cells with a duplicated *VSG* present in two ESs are often subsequently lost from the population or modified.

## Switched populations expressing MC-derived *VSGs* display a competitive advantage

To investigate VSG dynamics in switched populations, we next turned to analysis of the full 20-day time-course, using RNA-seq to quantify those switched variants present in the population. A principal component analysis, monitoring read-counts for the activated set of *VSGs*, showed that the population was dynamic, and also that the three technical replicates followed a similar and remarkably consistent path (Fig 3A). A striking pattern emerged when we compared the two major groups of *VSGs*; cells expressing *VSGs* derived from ESs were relatively diminished in abundance over time, while *VSGs* derived from minichromosomes increased in abundance over time; again, the behavior of the three populations was remarkably consistent (Fig 3B). Thus, the transition from ES-VSGs to MC-VSGs, previously thought to be driven by the adaptive immune response, in fact occurs in the absence of an adaptive immune response. Notably, these findings are consistent with our observation above regarding the diversity of VSGs in ESs, which suggested that cells with a duplicated *VSG* present in two ESs are subsequently lost from the population or modified. An analysis of individual *VSGs* from each group during the time-course revealed the underlying source of these trends in more detail. Indeed, ten of thirteen *VSGs* that initially diminished in abundance were derived from ESs; $\chi^2$ test $p = 0.005$ (Fig 3C).

To determine whether competition among variants, and the competitive advantage displayed by populations of cells expressing MC-*VSGs*, were reproducible, we generated another independent 'sgRNA break-site 3' strain and again induced VSG switching as described above. RNA-seq analysis was carried out using triplicate switched populations, from which mRNA samples were prepared on day-5 and on day-25 after switching was induced. A comparison of read-counts for the activated set of *VSGs* on day 5 revealed remarkably similar activation

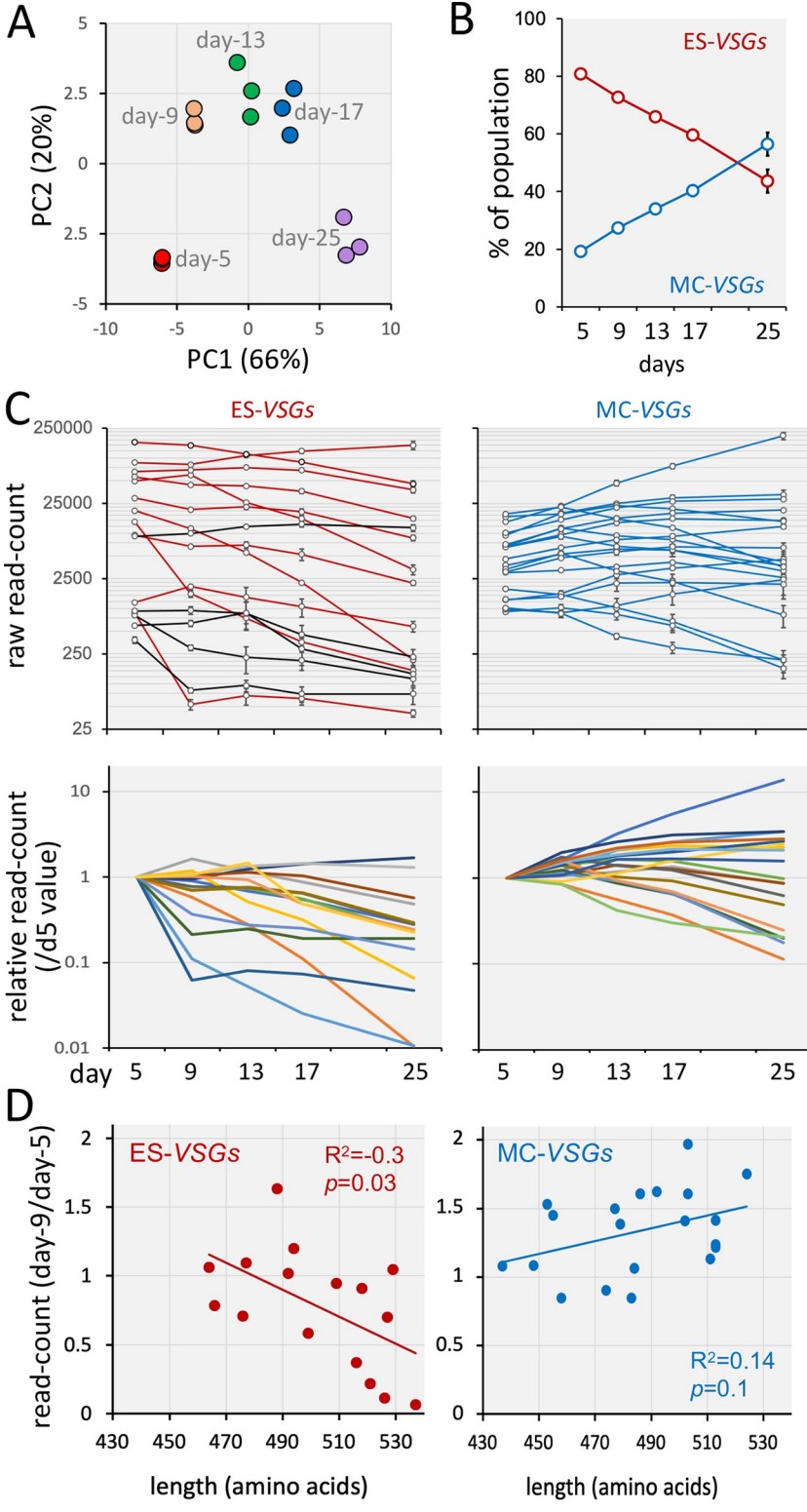

**Fig 3. VSG dynamics are related to VSG template location and VSG length.** (A) A principal component analysis of read-counts (S1 Data) for the activated *VSGs* (n = 36) in each replicate during the RNA-seq time-course. (B) Relative read-counts for ES-*VSGs* (n = 16) and MC-*VSGs* (n = 20) over the RNA-seq time-course. Error bars, SD; three replicates. (C) Read-counts for individual ES-*VSGs* and MC-*VSGs* over the RNA-seq time-course. The upper panels show bloodstream ES-*VSGs*, red; metacyclic ES-VSGs, black; MC-*VSGs*, blue. Error bars, SD; three replicates. The

lower panels show the same dataset but displayed relative to the day-5 values to emphasize the different behavior for the two sets of VSGs. (D) The graphs show fold-change in read-counts for either ES-*VSGs* or MC-*VSGs* between the day-5 and day-9 time-points relative to VSG length. R$_2$ and *p* values were derived using regression analysis in Excel.

frequencies for individual *VSGs* ($R^2 = 0.47$; $p = 4^{-6}$), those derived from both expression-sites and minichromosomes (S3 Fig). A comparison of read-counts over the 20-day timeframe also revealed a remarkably similar pattern of relative growth for populations expressing individual *VSGs* ($R^2 = 0.89$; $p = 8^{-18}$), again including those derived from both expression-sites and mini-chromosomes (S3 Fig).

Given the similar activation rates and subsequent growth rates observed above, we suspected that analysis of the second independent 'sgRNA break-site 3' strain would also reveal diminished abundance over time for cells expressing *VSGs* derived from ESs and increased abundance over time for *VSGs* derived from minichromosomes, as observed in Fig 3B. This was indeed the case, and once again, the behavior of the three replicate populations was remarkably consistent (S3 Fig). An analysis of individual *VSGs* from each group revealed the underlying source of these trends in more detail (S3 Fig). We concluded that competition among variants, and the competitive advantage displayed by populations of cells expressing minichromosome derived *VSGs*, were highly reproducible in independent biological replicates.

To determine whether growth differences could also be observed in isolated clones, we compared a switched clone expressing an ES-*VSG* (*VSG-3*) and three switched clones expressing MC-*VSGs*; two expressing *VSG-1* and one expressing *VSG-23*. Cells were seeded at $10^5$ / ml in duplicate, split each day, and counted after three days. Doubling times were 6.3 h, except for one of the *VSG-1* expressing clones, which displayed a doubling time of 6.5 h (S4 Fig). We next mixed *VSG-3* expressing cells 50:50 with each of the other clones and counted the proportion of VSG-3 negative cells by microscopy over two weeks of competitive growth. In all three cases, the *VSG-3* negative, MC-*VSG* expressing cells increased to 70–75% of the population, within only 3-days in three cases (S4 Fig). These results suggest "that the growth rate of clones of trypanosomes grown individually may not correlate with their relative growth rates when grown in mixtures", as stated more than forty years ago [30], and as also observed in an even earlier independent study [22].

## VSG dynamics in switched populations transiently correlate with VSG length

We were interested in the possibility that trypanosomes expressing different-length VSGs grow at different rates [19]. To determine whether there was a relationship between *VSG* expression dynamics and *VSG*-length, we assessed changes in *VSG* abundance at the earliest time-points. When considering the full set of thirty-six *VSGs*, we saw no significant correlation between length and abundance changes ($p = 0.1$); perhaps unsurprising given the distinct behavior of ES-*VSGs* and MC-*VSGs* (Fig 3B and 3C). In contrast, we saw an inverse and significant correlation between length and abundance changes for ES-*VSGs* ($R^2 = -0.3$; $p = 0.03$) and a weak correlation between length and abundance changes for MC-*VSGs* ($R^2 = 0.14$; $p = 0.1$) (Fig 3D). Other parameters failed to achieve a significant correlation with VSG abundance changes for either cohort of *VSGs*; including VSG mass (Fig 4A), number of predicted N-glycosylation sites per VSG (Fig 4B) or number of Cys residues per VSG (Fig 4C). VSG activation frequency (Fig 4D) and VSG abundance changes between day-9 and day-13 (Fig 4E) also failed to achieve a significant correlation with VSG length for either cohort of VSGs. These results

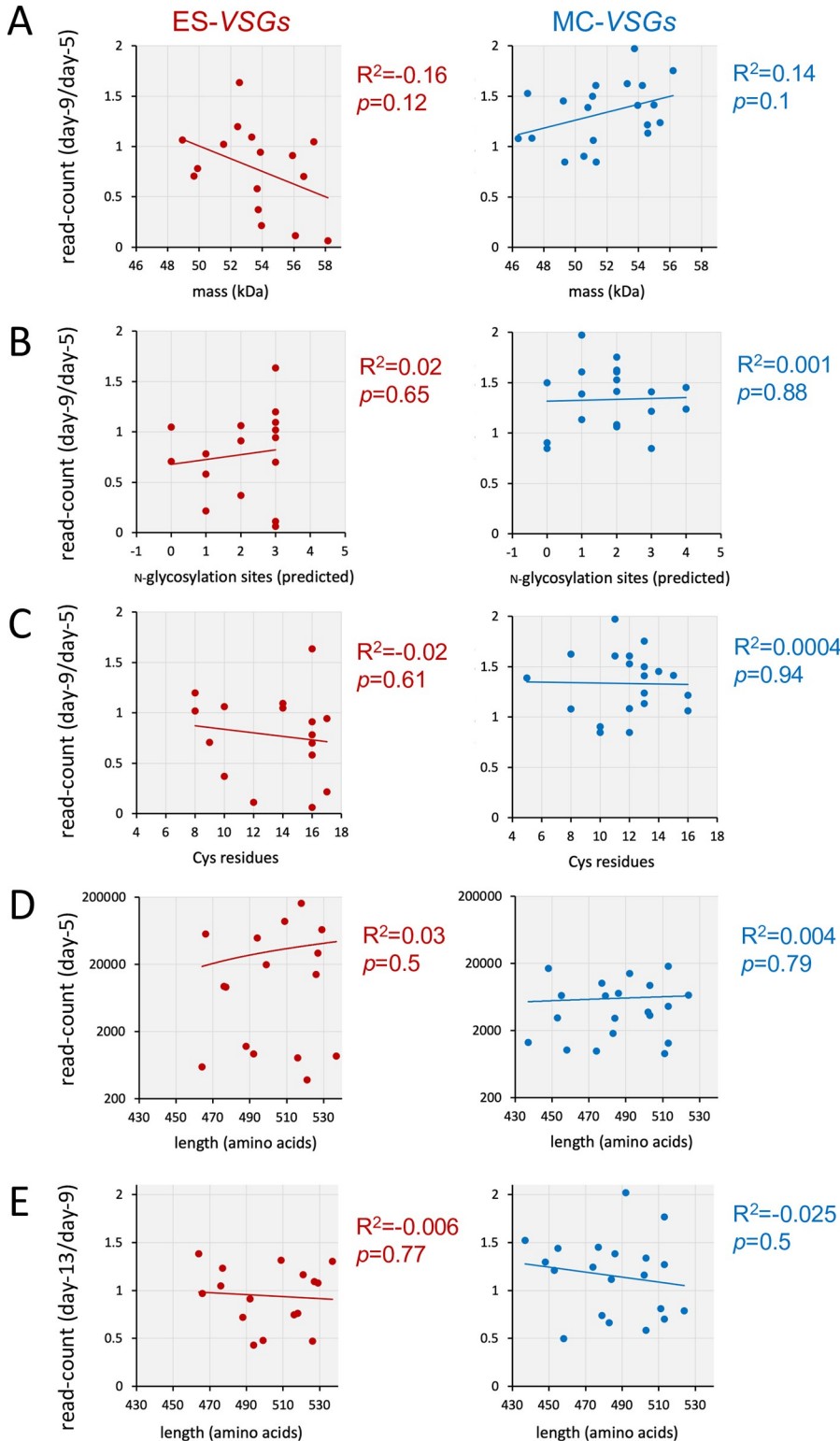

**Fig 4. VSG dynamics and relationship to other features of VSGs.** (A) The graphs show fold-change in read-counts for *VSGs* between the day-5 and day-9 time-points relative to VSG mass. (B) As in A but relative to predicted N-glycosylation sites. (C) As in A but relative to number of Cys residues. (D) The graphs show read-counts for *VSGs* at the day-5 time-point (a measure of VSG activation rate) relative to VSG length. (E) The graphs show fold-change in read-counts for *VSGs* between the day-9 and day-13 time-points relative to VSG length. Further comparisons between

the day-13/17 and day-17/25 time-points and VSG length yielded $R_2$ values <0.1 and $p$ values >0.3. All $R_2$ and $p$ values were derived using regression analysis in Excel.

suggest that the length of the expressed *VSG* does indeed impact trypanosome growth rate, as suggested previously [19], based on an analysis of VSG population dynamics *in vivo* [5].

## Differential growth is associated with wider transcriptome differences

We next asked whether contemporary switched cells growing at different rates displayed transcriptome differences that extended beyond the active *VSG*. A principal component analysis, monitoring the transcriptome over the 20-day time-course, again revealed a dynamic population, and also a remarkably consistent path for the three technical replicates (Fig 5A). Notably, the wider transcriptome appeared to diverge only transiently. A comparison of RNA-seq data from the day-5 samples relative to either the day-9 or day-25 samples confirmed this view (Fig 5B) and revealed significant differences in gene expression at earlier time-points, as highlighted by transiently reduced expression of genes encoding nucleolar proteins, tRNA synthetases and glycolytic enzymes (Fig 5B). These results supported the hypothesis that differential growth is indeed associated with wider changes in the transcriptome.

To further explore this hypothesis, we generated a panel of switched clones by once again inducing DNA breaks at sgRNA site-3 (see Fig 1B). Among thirty-six independent clones analyzed, we identified clones expressing several different ES-*VSGs* (VSG-3, 6, 8, 9, 11, 13, 17, 631) and MC-*VSGs* (VSG-24, 25, 476, 832, 3039), as determined by *VSG*-generic RT-PCR and Sanger sequencing (see Fig 5C). A comparison with results shown in Fig 2D suggests >30% under-sampling of clones that activate ES-*VSGs* here, potentially reflecting declining growth-rates that are incompatible with clone expansion in some cases. Since the changes we sought to characterize may be transient and/or may result in the loss of clones that activate ES-*VSGs*, we assessed growth rates at the earliest time-point possible; 5 days after subcloning. The analysis revealed different growth rates associated with different VSGs (Fig 5C), and based on the results, six clones were selected for RNA-seq analysis; four clones that activated ES-*VSGs*, two slower-growing (VSG-6 or VSG-8) and two faster-growing (VSG-9 or VSG-11), and two faster-growing clones that activated MC-*VSGs* (VSG-25 or VSG-832). The relative growth of these clones is shown in Fig 5C (right-hand panel).

RNA-seq analysis confirmed loss of *VSG-2* (no reads mapped) and exclusive expression of the expected new *VSG* by each clone (av. 99.6% of VSG reads mapped). A principal component analysis, comparing the wider transcriptomes of these clones, revealed clustering of the faster-growing clones, while the pair of slower-growing clones were outliers (Fig 5D). Indeed, the slower-growing clones displayed divergent transcriptome profiles, that correlated with growth-rate, and also the signatures described above, characterized by negative correlation between growth-rate and relative expression of nucleolar proteins, glycolytic enzymes and tRNA synthetases (Fig 5E). These results support the view that activation of an ES-*VSG*, when compared to activation of a minichromosomal *VSG*, often results in relatively slow growth that is correlated with transcriptome remodeling.

## Discussion

African trypanosomes are remarkably persistent parasites, which can maintain infections in the bloodstream of immune-competent hosts for years. In experimental *T. brucei* infections, a VSG hierarchy is observed, which can prolong infection, but how this arises is not fully understood. Although competition among VSG variants was reported more than forty years ago [22,

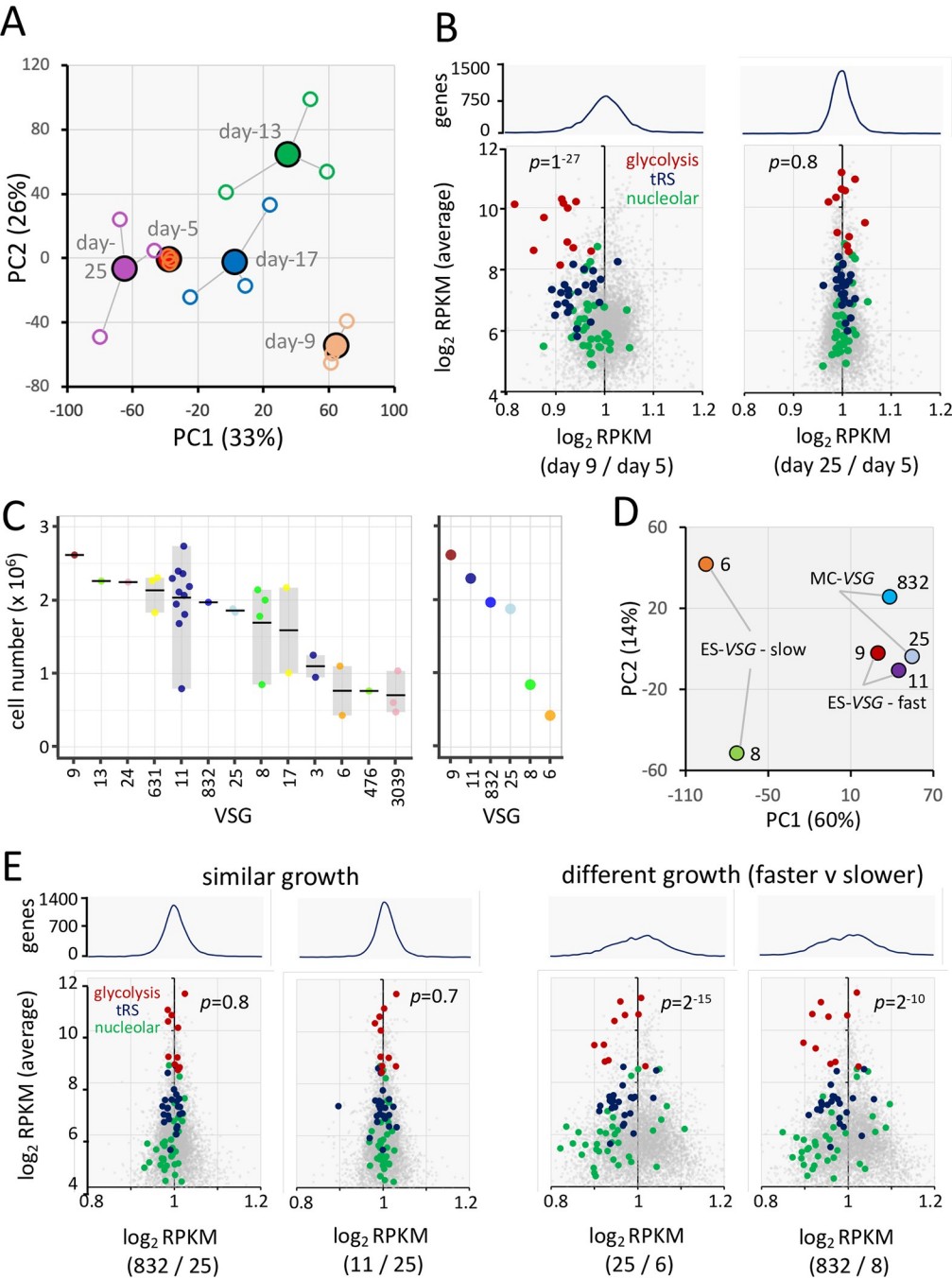

**Fig 5. Differential growth is associated with wider transcriptome differences.** (A) A principal component analysis of $\log_2$ RPKM (reads per kilobase per million) values (S1 Data) for the transcriptome (n = 7255) during the RNA-seq time-course. The closed datapoints represent averages while the open datapoints represent each replicate. (B) The scatterplots show transcriptome differences between the day-5 and either day-9 or day-25 samples, highlighting transcripts encoding nucleolar proteins (n = 35), tRNA synthetases (n = 23) or glycolytic enzymes (n = 11). $p$ values were derived using a $\chi^2$ test for increased v decreased expression of these 69 genes. The upper panels indicate data distribution. (C) The graphs show cell numbers for 36 clones and six of those clones selected for RNA-seq (right-hand panel), also indicating the VSG expressed by each clone. The data are shown as jittered dots and, when n>1, the horizontal line indicates the mean and the box indicates the range of values. (D) A principal component analysis of $\log_2$ RPKM values (S1 Data) for the transcriptomes of the six selected clones. (E) The scatterplots show transcriptome differences between the pairs of clones indicated. Other details as in B above.

30], more recent studies have focused on switch-rates rather than subsequent growth rates. One recent study did report changes in the average length of VSGs expressed in a mouse model, however, pointing to a role in differential growth [19]. Our goal was to experimentally assess factors that contribute to the VSG hierarchy, specifically to assess the hypothesis that cells expressing different VSGs compete. Using inducible Cas9 to introduce programmed DNA breaks at the active *VSG*-ES, we generated a highly heterogeneous population of cells expressing different VSGs. VSG dynamics were then monitored using RNA-seq. We show that variants do indeed compete. Thus, parasite-intrinsic factors that drive population structure are more sophisticated than previously appreciated.

Our Cas9-based approach allowed us to generate a complex population of cells expressing different VSGs at the telomeric end of the same single-copy expression site on chromosome 6a. RNA-seq analysis then provided a measure of relative VSG activation frequency, at the earliest post-switching time-point, and relative growth rate, at subsequent time-points. We detected activation of sixteen distinct ES-*VSGs* and twenty distinct MC-*VSGs* which could be effectively monitored during the time-course. In terms of relative VSG activation frequency, we were able to monitor three distinct groups of *VSGs*; those within polycistronic ESs, which typically have long tracts of 70-bp repeats [8]; those within monocistronic ESs [40], which typically have short tracts of 70-bp repeats; and minichromosomal or MC-*VSGs* [38], which also typically have short tracts of 70-bp repeats [42]. Although the length of 70-bp repeat tracts remains unknown at many sites, we found that the frequency of *VSG* activation was broadly consistent with the previously reported lengths of these tracts [8,38,40,42]. Indeed, VSG-14 and VSG-15, the polycistronic ES-associated *VSGs* reported to have the shortest adjacent tracts of 70-bp repeats [8], were activated at a lower frequency than any other polycistronic ES-associated *VSG*. Thus, recombination efficiency appears to be dependent on the length of 70-bp repeat substrate available; as is also the case for homologous recombination amongst other *T. brucei* sequences [43]. Mathematical modelling indicated that VSG length-dependent activation rates have the potential to allow *T. brucei* to persist for moderately longer *in vivo* [19]. Our findings, however, do not provide support for VSG length-dependent activation rates.

Our time-course analysis revealed competition among *T. brucei* cells expressing VSGs of different lengths. The set of VSGs analyzed here ranged from 437 to 537 amino acids. Relatively few of the annotated VSGs have been shown to be functional, but a VSGnome collection comprises 325 apparently intact genes ranging in length from 436–582 amino acids [38]. This 33%, 146 amino acids differential suggests that the capacity for VSG-length dependent differential growth may be greater than sampled here. We suggest that differential VSG activation-frequency and differential growth both contribute to prolonging persistence *in vivo*; VSG flanking homology (70-bp repeats in particular) primarily determines the activation frequency while VSG template location, and also VSG length, determine subsequent growth rate. Further transcriptome analysis revealed a signature associated with differential growth. Specifically, recently switched, and slower-growing trypanosomes expressing ES-*VSGs* displayed increased expression of genes involved in nucleolar function, protein translation and energy metabolism.

Why do cells that activate ES-*VSGs* grow slower than cells that activate MC-*VSGs*? We suggest a mechanism based on homology-dependent interference, proposed previously as a mechanism that promotes allelic exclusion and transcriptional repression [44]. An active *VSG* that is identical to the repressed ES-located VSG from which it was copied may itself be subject to competition and interference. Indeed, all ES-located VSGs appear to be unique [8], supporting the view that cells with pairs of identical ES-associated *VSGs*, although they may arise often, are typically lost from the population, or modified. This feature may also present a potential route for assembling mosaic VSGs, recognized as important for sustaining more persistent

infections. It has remained unclear how mosaic VSGs are assembled through multiple recombination steps without disrupting VSG function; especially given the high proportion of *VSG* pseudogenes and gene fragments present in the genome [45]. Our findings now suggest a solution to this conundrum. Cells that activate ES-*VSGs* may escape immune surveillance due to homology-dependent interference and slower growth, as detailed above. These populations may then circulate at low density for sufficient periods of time to allow for the assembly of mosaic VSGs. Cells expressing defective mosaic VSGs would be rapidly eliminated, since continuous VSG expression is required for proliferation and viability [46], while functional mosaics would support continued growth. Notably, mosaic formation would also relax the homology-dependent interference proposed above. Our findings can also explain why the average ES-*VSG* is significantly longer than the average MC-*VSG*; because cells expressing longer MC-derived *VSGs* (that are not present in a second expression site) have a competitive advantage.

It has been known since the 1960s that VSGs emerge in a hierarchical order during a *T. brucei* infection [21,23], but our understanding of the mechanisms underpinning this behavior has remained incomplete. We now show that the activated VSGs themselves, their locus of origin in the genome and their length, can make important contributions to the hierarchy, thereby supporting a more persistent infection than would otherwise be possible [19]. Notably, competition among variants was reported more than forty years ago [22,30], and we report similar competition here. In particular, we find that populations of trypanosomes that activate MC-*VSGs* exhibit a competitive advantage, highlighting the importance of the minichromosomal repertoire for establishing an infection. Thus, we suggest that a complex set of VSG variants emerges, but that relatively few achieve a sufficient density to trigger an effective immune response. This represents a feint attack strategy [19], whereby *T. brucei* diverts the host acquired immune response with faster-growing variants, allowing slower-growing variants to persist before eventually emerging from below-threshold sub-populations. Such an economical use of VSGs allows *T. brucei* to stay one step ahead of the host immune response. These behaviors likely delay the exhaustion of available VSGs and help to explain why *T. brucei* is such a persistent parasite.

## Materials and methods

### *T. brucei* growth and manipulation

*T. brucei* 2T1-T7-Cas9 cells [33] were grown in HMI-11 medium at 37˚C and 5% $CO_2$, with hygromycin (1 μg/mL) and blasticidin (2 μg/mL). Transfections were performed using electroporation. sgRNA expression cassettes 1–3 were introduced into *T. brucei* as described [33]; in the absence of tetracycline. In this case, transformed clones were selected in hygromycin, blasticidin and phleomycin (2 μg/mL).

### Plasmid construction

The sgRNA constructs 1–3 were assembled as described [33] and using the following oligonucleotides: gRNA1fw (AGGGATGGATTTGAAGTGATGATG), gRNA1rv (AAACCATCATC ACTTCAAATCCAT); gRNA2fw (AGGGAGAATTCGGAGACGAAGAGC), gRNA2rv (AAA CGCTCTTCGTCTCCGAATTCT); gRNA3fw (AGGGTGCCTTCCAATCAGGAGGCC), gRNA3rv (AAACGGCCTCCTGATTGGAAGGCA). Correct assembly was confirmed by sequencing.

## Western blotting

Protein blotting was carried out according to standard protocols. PBS-T-M (PBS, 0.1% Tween-20, 5% milk powder) was used to block the membrane for 1 h. The >70 kDa part of the membrane was used for Cas9 detection with a mouse α-Cas9 primary antibody (1:1000, Abcam) and α-mouse HRP secondary antibody (1:2000, BioRad), and the <70 kDa part of the membrane was used for EF1α detection with a mouse α-EF1α primary antibody (1:20000, Merck Millipore) and an α-mouse HRP secondary antibody (1:2000, BioRad). Primary antibodies were incubated at 4°C overnight and secondary antibodies were incubated for 1 h at RT. Membranes were developed using ECL solution (GE healthcare) and images were acquired using a ChemiDoc XRS+ (BioRad).

## Immunofluorescence microscopy

Cells were washed in PBS and fixed in 3% formaldehyde in PBS at 37°C for 10 min, plus 5 min at RT. Following washing in PBS, 1% BSA and $H_2O$, the cells were resuspended in 30 μL 1% BSA. 5 μL of a suspension was placed on each well of a 12-well, 5 mm slide (Thermo scientific) and dried overnight at RT. Cells were then re-hydrated for 5 min in PBS and blocked for 10 min with 50% FBS. Following two washes in PBS, cells were incubated in primary antibody for 1 h at RT; rat α-VSG-2 (1:10,000), rabbit α-VSG-6 (1:10,000), mouse α-VSG-3 coupled with Alexa 488 (1:500). Following three washes in PBS, cells were incubated in secondary antibody for 1 h at RT; α-rat rhodamine (1:2000 Pierce), α-rabbit Alexa 488 (1:2000, Life Technologies). Cells were washed three more times in PBS and mounted in medium containing DAPI (Vecta-Shield). Microscopy was performed using a Zeiss Axiovert 200M inverted light microscope with a magnification of 10 x 60 with oil immersion. Images were acquired using ZenPro (Carl Zeiss, Germany) and the same parameters and then equally processed using Fiji v1.5.2e.

## Cas9-induced antigenic variation

*T. brucei* 2T1-T7-Cas9 cells with the appropriate sgRNA, as described above, were induced to express Cas9 by the addition of tetracycline (1 μg/mL). For the RNA-seq time-course (see below), cells expressing sgRNA-3 were grown in tetracycline for 3-days (1 μg/mL) and then washed in HMI-11 to remove tetracycline. This culture was split into three parallel cultures on day 5 following tetracycline addition and the cells were subsequently split to $1 \times 10^4$ cells/mL in 50 mL every two days. Approximately 10 million cells were collected from each of the three flasks for RNA extraction using an RNeasy Mini Kit (Qiagen) on days 5, 9, 13, 17 and 25 following tetracycline addition, or on days 5 and 25 only for the second 'sgRNA-3' biological replicate; samples were taken for immunofluorescence microscopy in parallel. Following cell lysis, samples were stored at -80°C and further processed in parallel. Individual switched clones were generated by inducing *T. brucei* 2T1-T7-Cas9 cells with sgRNA-3 as above. Following tetracycline removal, the cells were allowed to recover for 2-days, and cells were then subcloned by limiting dilution in 500 ul of HMI-11 in 48 well-plates. Five days after subcloning 100 μL of medium was collected for each clone and VSG-2 expression was assessed by immunofluorescence microscopy. The number of parasites in each well for those clones that no longer expressed VSG-2 was quantified using a CASY Cell Counter (Cambridge Bioscience). These clones were grown for a further 2-days for RNA extraction as described above. Additional clones were selected for growth analysis in isolation, and in competition, using immunofluorescence microscopy.

## RNA sequencing and analysis

Briefly, for RNA-seq, polyadenylated transcripts were enriched using polydT beads and reverse-transcribed. For the time course, sequencing was on a HiSeq 4000 platform (Illumina): approximately 40 million 100 b paired-end reads were generated per sample (BGI, Hong-Kong). For analysis of switched clones, and for analysis of the second 'sgRNA-3' biological replicate, sequencing was on a DNBSEQ-G400 platform (MGI): approximately 70 million 100 b paired-end reads were generated per sample (BGI, Hong-Kong). The forward and reverse paired-end reads were aligned to the *T. brucei* TREU927 reference genome, v55 downloaded from TriTrypDB [47], and to the set of *VSG*-ESs from the *T. brucei* Lister 427 strain [8], using bowtie2 [48], with the 'very-sensitive-local' pre-set alignment option. The alignments were converted to BAM format, reference sorted and indexed with SAMtools [49]. Fragment counts were determined from the BAM files using featureCounts [50] with parameters: -p (pair end) -B (both ends successfully aligned) -C (skip fragments that have their two ends aligned to different chromosome) -M (count multi-mapping) -O (match overlapping features) -t CDS (count level) -g gene_id (summarization level). Genes with low counts were filtered out using edgeR [51] and RPKM values were extracted with cqn.

The alignments to the set of '1,200 bp' truncated *VSGs* were filtered for MapQ > 1 with SAMtools. We considered a set of ES-*VSG* sequences, and a set of MC-*VSG* sequences [8,37,38,40], and only considered these *VSGs* to be activated if they registered >10-fold the proportion of *VSG* reads typically observed for silent *VSGs*; these 'background' values were 0.004% for B-ES *VSGs*, 0.002% for M-ES *VSGs*, and 0.00003% for MC-*VSGs*. Average proportions for activated *VSGs* in the day-5 samples were 2.8% for B-ES *VSGs*, 0.15% for M-ES *VSGs*, and 0.37% for MC-*VSGs*; 671-, 65- and 14,195-fold above 'background', respectively. Additional *VSGs* in preliminary analyses included subtelomeric array *VSGs* (*VSG-4* and *VSG-5*), but these *VSGs* failed to register sufficient reads to qualify as activated. The scripts to align the RNA-seq data and to compute read-counts have been deposited in GitHub and Zenodo. To generate base-pair resolution plots, reads per kilobase per million were extracted using deepTools [52], plotted on Prism9 and then annotated and edited in Illustrator. Sanger sequencing of *VSG* mRNA was carried out in-house. Total RNA samples (1 μg) were used for first strand synthesis using a primer that anneals to all *VSG* 3'-UTRs (DH3, GACTAGTGTTAAAATA-TATCA) and M-MLV reverse transcriptase (Promega). *VSG* cDNA was then generated by PCR using hot start Taq DNA polymerase (NEB), DH3 primer, and a primer for the spliced leader sequence (DH5, GACTAGTTTCTGTACTATAT). PCR steps were five cycles of 30 s at 94˚C, 30 s at 50˚C, and 1 min at 72˚C, followed by 13 cycles of 30 s at 94˚C, 30 s at 55˚C, and 1 min at 72˚C. PCR products were resolved on agarose gels and cDNA was extracted using a QIAquick Gel Extraction Kit (Qiagen). *VSG* sequencing was performed using Applied Biosystems Big-Dye Ver 3.1 chemistry on an Applied Biosystems model 3730 automated capillary DNA sequencer.

## VSG annotation

Signal peptides were predicted using the SignalP-6.0 tool at https://services.healthtech.dtu.dk/service.php?SignalP or earlier iterations (SignalP-5.0, or SignalP-3.0). GPI modification sites were predicted using the big-PI predictor at https://mendel.imp.ac.at/gpi/gpi_server.html. N-glycosylation sites were predicted using NetNGlyc at https://services.healthtech.dtu.dk/service.php?NetNGlyc-1.0. Phylogenetic analysis was carried out using ETE 3 at https://www.genome.jp/tools-bin/ete [53].

## Supporting information

**S1 Fig. Filtering of the minichromosomal *VSG* set.** The VSG set was filtered based on RNA-seq read-mapping profiles. *VSGs* displaying full coding sequence activation were included in the analysis. *VSGs* with mapped reads restricted to the *C*-terminal coding sequence were excluded from further analysis. These results are explained by the presence of common sequences in multiple VSGs. The 3'-terminal 268 nucleotides of VSG-444 are shared with VSG-18, while the 3'-terminal 584 nucleotides of VSG-514 are shared with VSG-17, for example. VSGs included in the analysis were truncated to 1,200 bp to remove shared sequence.
(TIF)

**S2 Fig. Annotation and phylogenetic analysis of the activated VSG set.** (A) The schematic shows the activated set of VSG proteins with salient features highlighted. Signal peptides were predicted using the SignalP tool. GPI modification sites were predicted using the big-PI predictor. N-glycosylation sites were predicted using NetNGlyc. O-glycosylation sites are from [32]. VSG domain types are indicated. (B) The phylogenetic tree shows the activated set of VSGs analyzed using ETE 3 [53]; the *N*-terminal 400 amino acids only. (C) As in B but for the *C*-terminal 100 amino acids only. An asterisk indicates undefined VSG *C*-terminal type.
(TIF)

**S3 Fig. Reproducibility of VSG activation and MC-*VSG* associated competitive advantage.** (A) The graph shows *VSG* expression levels as determined by RNA-seq read-count on day 5 after inducing switching in two independent biological replicate strains. n = 36 *VSGs*. (B) The graph shows relative *VSG* expression levels as determined by relative RNA-seq read-count over 20 days of growth following switching in two independent biological replicate strains. n = 36 *VSGs*. (C) Relative read-counts for ES-*VSGs* (n = 16) and MC-*VSGs* (n = 20) at day-5 and day-25 in the second replicate strain, and as determined by RNA-seq; three replicates, error bars, SD (not visible). (D) Read-counts for individual ES-*VSGs* (n = 16) and MC-*VSGs* (n = 20) at day-25 relative to the day-5 values in the second replicate strain.
(TIF)

**S4 Fig. Growth and competitive growth of individual clones.** (A) The graph shows doubling times calculated for individual clones expressing the VSGs indicated. Duplicate flasks were counted after three days of growth; quadruplicate flasks in the case of the VSG-3 expressing clone. The horizontal bars indicate average values. (B) The *VSG-3* expressing cells were mixed 50:50 with each of the other clones shown in A. Cells were then stained with αVSG-3 on the days indicated, and the proportion of VSG-3 negative cells were counted by microscopy.
(TIF)

**S1 Data. RNA-seq data.** Read-counts and RPKM values for the time course, for the set of six switched clones, and for the second replicate clone. Excel format.
(XLSX)

## Acknowledgments

We thank E. Rico for advice on use of the inducible Cas9 system; S. Duncan and M. Ferguson for advice on N-glycosylation and S. Hutchinson (Institut Pasteur, Paris), N. Siegel and R. Cosentino (LMU, Munich) for advice on VSG sequences. We also thank the Sequencing & Services team (MRC PPU, School of Life Sciences, University of Dundee, Scotland).

## Author Contributions

**Conceptualization:** David Horn.

**Data curation:** Michele Tinti, Joana Faria, David Horn.

**Formal analysis:** Douglas O. Escrivani, Viktor Scheidt, Michele Tinti, Joana Faria.

**Funding acquisition:** David Horn.

**Investigation:** Douglas O. Escrivani, Viktor Scheidt.

**Methodology:** Douglas O. Escrivani, Viktor Scheidt.

**Supervision:** David Horn.

**Visualization:** Michele Tinti, Joana Faria.

**Writing – original draft:** David Horn.

**Writing – review & editing:** Douglas O. Escrivani, Viktor Scheidt, Michele Tinti, David Horn.

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
