## [Decision Letter · Decision Letter 0]

7 Jan 2023

Dear Prof. Horn,

Thank you very much for submitting your manuscript "Competition among variants is predictable and contributes to the antigenic variation dynamics of African trypanosomes" for consideration at PLOS Pathogens. As with all papers reviewed by the journal, your manuscript was reviewed by members of the editorial board and by several independent reviewers. In light of the reviews (below this email), we would like to invite the resubmission of a significantly-revised version that takes into account the reviewers' comments.

We cannot make any decision about publication until we have seen the revised manuscript and your response to the reviewers' comments. Your revised manuscript is also likely to be sent to reviewers for further evaluation.

Sincerely,

F. Nina Papavasiliou

Guest Editor

PLOS Pathogens

Margaret Phillips

Section Editor

PLOS Pathogens

Kasturi Haldar

Editor-in-Chief

PLOS Pathogens

orcid.org/0000-0001-5065-158X

Michael Malim

Editor-in-Chief

PLOS Pathogens

orcid.org/0000-0002-7699-2064

Reviewer's Responses to Questions

**Part I - Summary**

Reviewer #1: In this paper, Scheidt and Escrivani et al demonstrate that T brucei populations composed of parasites expressing multiple VSGs show predictable patterns of VSG expression in vitro over time, in the absence of a host immune response. They find that initially expression-site associated VSGs emerge, but later minichromosomal VSGs establish within the populations. There is a slight correlation between parasite fitness/outgrowth and VSG length, and slow-growing parasite clones show transcriptional differences compared to faster ones. Overall, their data support the idea that competition between different VSG-expressing subpopulations plays a role in this parasite's ability to evade host adaptive immunity for very long periods of time, which is novel and interesting. The paper would be greatly improved by the use of biological replicates (multiple parasite clones and multiple break inductions) to support their findings.

Reviewer #2: In this work, the authors use CRISPR technology and genome-wide RNA sequencing to tackle an old question in the antigenic variation field: whether post-VSG switching populations compete with each other, thus contributing to the global predicted pattern of VSG expression during infection. The authors characterized the dynamics of the heterogenous VSG-switched population over time (day 5 to 25 post-VSG switch). Parasites expressing VSG from Mini-chromosomes became predominant in the switched population, while VSG from Expression Sites (ES) became rarer. This effect was associated with the fact that the ES-VSG were encoded by a longer gene. More careful analysis of the full transcriptome revealed that parasites that activate long ES-VSG undergo a transient reduction in growth (day 9), which is no longer detectable at day 25. The authors do not study how expression of a longer VSG could affect growth rate, but they propose that such reduced transient growth could be beneficial for not triggering a strong host immune system immediately after VSG switching.

This works has an original angle on antigenic variation, because it focuses on the population dynamics after switching instead of studying the switching moment or steady-state conditions where VSGs are maintained silent/active. The paper is very well written and the figures are extremely clear.

Reviewer #3: This manuscript by Scheidt and colleagues provides an analysis of the parameters that affect activation and/or abundance of Variant Surface Glycoprotein (VSG) during antigenic variation by Trypanosoma brucei. The approach taken is to model a switch in expressed VSG by conditional expression of Cas9 endonuclease, targeting a DNA double strand break around the transcribed VSG gene within the bloodstream VSG expression site. While the parameters affecting the efficiency of VSG selection/expression are thoroughly and clearly explored, I’m afraid that I find the extensive discussion and conclusions reached by the authors overreach the constraints of the assay system adopted.

The two major conclusions reached are that VSG length dictates VSG activation efficiency or abundance (it is unclear which they favour), and that differential growth rates of parasites newly expressing distinct VSGs is reflected in transcriptome differences. As explained below, both conclusions are insecurely grounded in the available data. In addition, in the discussion, the authors make further claims from their experiments that have not been tested adequately.

**Part II – Major Issues: Key Experiments Required for Acceptance**

Reviewer #1: - Were all the experiments performed using one sgRNA clone? From the text this appears to be the case. If so, one cannot rule out (however unlikely) that the patterns observed are clone-specific. it would be good to know that these patterns transcend the individual clone. At the very least it would be important to acknowledge the possibility that individual clones may have different intrinsic tendencies.

- Similarly, using technical replicates for the RNA-seq, instead of three separate break inductions, might reveal an induction-specific set of outcomes that wouldn't occur every time a cut is made, even though the three parallel cultures follow the same trajectory. The fact that the second induction of a break (data in figure 5C) resulted in a slightly different set of clones, further raises suspicion that this pattern might be more variable than the initial experiment suggests. The authors state that this result reflects under-sampling, possibly due to growth differences influencing clonal expansion, but without parallel RNA-seq data it is hard to be sure this is truly under sampling and not a slightly different experimental outcome.

Reviewer #2: Most conclusions are well supported by the findings. I have three questions related to Figure 5, from which authors conclude that growth rate appears to be transiently reduced when parasites switch from VSG-2 to a longer VSG (ES-VSGs).

1. On day 25, the authors no longer detect the temporary reduction of transcript levels related to energy metabolism and translation first detected in day 9, suggesting that on day 25 all switchers proliferate at similar rates. This result should be directly demonstrated by either measuring growth rate of individual subclones isolated on day 25 (expressing different VSGs, of different chromosomal locations and lengths) or, even better, by doing competition experiments between such subclones.

2. It is not clear how growth rate was measured from the RNAseq data. Typically the units for cell proliferation are in time/division. While the evidence of a temporary reduction of transcript levels related to energy metabolism and translation is compelling and consistent with a transient reduced growth rate, the authors’ conclusion on transient reduced growth rate would be more robust if they could directly measure proliferation rate of individual switchers over time. It seems this could be achieved by flow cytometry analysis of parasite population post-switching using VSG antibodies (VSG-9, VSG-6, VSG-13, VSG-3) and Proliferation dies (such as Cell Trace Violet) at multiple days post-induction.

3. I wonder if the transient reduced growth rate is because parasites need to adjust from translating/expressing a small VSG (VSG-2) to a large VSG. Would the same transient reduced growth rate be observed if the starting parasite population expressed a large VSG (such as VSG-6 or VSG-8) instead of VSG-2, and switched to another large VSG? What would happen to proliferation if parasites switched from a large to a short VSG?

Reviewer #3: 1. The authors conclude that, amongst a range of factors that affect VSG abundance after induced switching (including genomic location and flanking sequence homology), is VSG coding sequence length. As stated above, the authors need to make clear what their RNA-seq approach provides a readout for: activation probability, or an effect of the expressed VSG on growth? For instance, in the discussion they state: ‘RNA-seq analysis then provided a measure of relative VSG activation frequency and relative subsequent growth rate for cells expressing different VSGs’. They also state: ‘Our findings, however, do not provide any support for VSG length-dependent activation rates’. Clearly these two processes are not equivalent and require differing hypotheses to explain the data. It seems the authors infer that RNA-seq at a single time point is a measure of activation, while comparison of two time points is a measure of relative growth, but it is not explained why these effects can be inferred in this way.

Irrespective, the data presented is contradictory for an influence of VSG length, and hence there seems to be no compelling case for such a simple relationship:

Fig.3D presents the main evidence for this conclusion but reveals an opposite correlation between abundance and length of VSGs found in silent expression sites (shorter = greater abundance) and in minichromosomes (longer = greater abundance). If VSG length is, as suggested, a primary determinant of VSG usage, the authors need to provide an explanation for why shorter VSGs in one location are activated/grow more efficiently, while longer VSGs are activated/grow more efficiently in another. The only explanation provided appears to make no sense: ‘What is less clear is why cells expressing longer minichromosome-derived VSGs appear to grow faster’. All VSGs are expressed from the same location (the VSG expression site), so why should the silent location from which they are sourced have any influence on growth, rather than activation efficiency? If VSG length is a determinant of VSG activation, why is this different between silent VSGs in the expression sites and in the minichromosomes?

A further complication, acknowledged by the authors in the results but ignored in the wider paper, is that relationship between VSG length and activation/growth efficiency (as detailed above and in Fig.3D) is only seen when comparing cells 9 and 5 days after Cas9 induction; Fig. 4E shows that no such length effect is seen when comparing days 9 and 13. This is not explained and, furthermore, there is not attempt to analyse the data across the full range of times that were sampled (up to days 17 and 25).

2. The authors suggest that the effect of the length of an expressed VSG on growth can be explained by transcriptome differences in cells expressing long and short VSGs: e.g. (abstract) ‘Differential growth of switched clones was also associated with wider transcriptome differences, affecting transcripts involved in nucleolar function, translation, and energy metabolism’. In the discussion, the authors present a more nuanced (and correct) argument: ‘recently switched, and slower-growing trypanosomes expressing ES-derived VSGs displayed increased expression of genes involved in nucleolar function, protein translation and energy metabolism’. Here again, the significance of this finding is unclear. It is not explained why expression of different length VSGs would impose differential levels of changes to these aspects of the transcriptome, or why this is ‘transient’. A possibility the authors appear not have not considered is that the transcriptome changes they detect at day 5 are due to the Cas9-induced break; either because some breaks are unrepaired, or lingering transcriptome effects of the cells responding to the break. The authors state ‘we assessed growth rates at the earliest time-point possible’; when was this, and have they tested the timing of Cas9-induced break repair? Again, such information is needed to separate activation from growth.

**Part III – Minor Issues: Editorial and Data Presentation Modifications**

Reviewer #1: VSG/RNA-seq mapping and analysis:

- Where did MC sequences come from? It seems like these are from the Cross "VSGnome" but the methods section only mentions the ES-VSGs and the 927 genome. This should be clarified.

- Similarly, were the only VSGs aligned to MC and ES VSGs? This potentially means that other (complete) VSGs in the genome could be coming up but aren't being evaluated by this analysis. If this is the case it should be noted somewhere.

- Why is the --very-sensitive-local option used for alignment instead of very sensitive in end-to-end mode? I know this has been shown to work well for BES mapping but I would worry that the soft clipping that (I think) is allowed in local mode could lead to incorrect mapping to some VSGs. Similarly, counting multi-mapping reads could lead to misleading results. Is there an advantage to counting this way? Since all comparisons are between multiple groups, and one VSG reference is used for all of the samples, it should be fine to count only uniquely mapping reads. It's possible there are not very many multi-mapping reads occurring in the analysis presented; this could also be reported.

- I can't figure out how this conclusion is drawn from the RNA-seq data : "We also note here that despite frequent duplicative conversion of ES-VSGs, all ESs appear to encode unique VSGs, suggesting that cells with a duplicated VSG present in two ESs are often subsequently replaced or modified." I had to read this sentence quite a few times to understand what it was getting at, so the authors might also consider rephrasing.

VSG Length:

- Correlation of survival/establishment with length seems potentially overstated. Its worth discussing but I would be hesitant to present this as a major finding given the relatively weak association, particularly for the MC- VSGs. The finding that MC VSGs preferentially establish is very interesting on its own and much stronger.

- How does the distribution of lengths of the MC VSGs detected in these populations compare to the distribution across the whole MC VSG repertoire?

MC VSG Emergence:

- Do the authors think that the MC VSGs are early switchers that are establishing later in the population or later switch events that establish after they appear in the population? It sounds like the former explanation is favored, but one could imagine that with a different background of competing variants (VSG-2 is not an option any longer) that some variants could be switched to at later timepoints and then establish. Similarly, maybe these are derived from the large VSG-6 population, and their emergence is related to the prior expression of VSG-6. Either way, I think the very interesting observation that " the transition from ES-VSGs to MC-VSGs, previously thought to be driven by the adaptive immune response, in fact occurs in the absence of an adaptive immune response" would still hold. It just might be useful to be more explicit about how precisely these variants might emerge.

Data Presentation:

- Figure 4 - There is a mistake in the figure- looks like a screenshot was taken while mousing over something?

- Figure 5b - What is being shown on the y axis? I assume RPKM for day 9/25, but it's not stated explicitly. Same for all of the plots of this type.

-There is no link to the github/zenodo sites with code

Reviewer #2: Minor comments:

• Could the authors explain how activation frequency was measured? RPKM / # days?

• Fig 3A (PCA based on VSG-expression). Could the authors speculate why parasite populations of day 25 are closer to day 5 (in component #2) than parasites on days 9, 13 and 17?

• The author summary is very clear!

Reviewer #3: 3. A large part of the discussion is concerned with untested contributions of 70 bp repeat length to VSG switching: ‘Although the length of 70-bp repeat tracts remains unknown at many sites, we found that the frequency of VSG activation was broadly consistent with the length of these tracts. Indeed, VSG-14 and VSG-15, the polycistronic ES-associated VSGs with the shortest adjacent tracts of 70-bp repeats [7], were activated at a lower frequency than any other polycistronic ES-associated VSG.’ Have the authors determined the lengths of the repeats in these loci in their cells?

4. Another focus of the discussion is mosaic VSG formation, but this seems rather a stretch since they have excluded all but full-length VSGs from their analysis. Can the authors provide any evidence that mosaic VSGs are present in the VSG expression sites after Cas9 switch induction?

PLOS authors have the option to publish the peer review history of their article (what does this mean?). If published, this will include your full peer review and any attached files.

Reviewer #1: No

Reviewer #2: **Yes: **Luisa M Figueiredo

Reviewer #3: No
---

## [Decision Letter · Decision Letter 1]

3 Jul 2023

Dear Dr. Horn,

We are pleased to inform you that your manuscript 'Competition among variants is predictable and contributes to the antigenic variation dynamics of African trypanosomes' has been provisionally accepted for publication in PLOS Pathogens.

Best regards,

F. Nina Papavasiliou

Academic Editor

PLOS Pathogens

Margaret Phillips

Section Editor

PLOS Pathogens

Kasturi Haldar

Editor-in-Chief

PLOS Pathogens

orcid.org/0000-0001-5065-158X

Michael Malim

Editor-in-Chief

PLOS Pathogens

orcid.org/0000-0002-7699-2064

Nothing to add beyond reviewer comments.

Reviewer Comments (if any, and for reference):

Reviewer's Responses to Questions

**Part I - Summary**

Reviewer #1: The manuscript is greatly improved by the edits and additional data, and the case for the authors' arguments is much more compelling now! I think is suitable for publication. I would recommend changing the p values in fig S3 and the text to normal scientific notation(2x10^-4 instead of 4^-6), however, for the sake of readability.

Reviewer #3: An interesting study, where the changes made have clarified the conclusions that can be reached. I note that all reviewers were concerned with the initial submission overstating the extent of correlation between VSG length and growth, and with confusion about how the assays distinguish activation frequency and growth. These aspects have been improved by careful rewording, addition of new data and further data analysis. I remain unconvinced that the study allows the authors to reach any conclusions about VSG 'mosaic' formation or use, but am happy to accept that it is clear from the wording that they are speculating.

**Part II – Major Issues: Key Experiments Required for Acceptance**

Reviewer #1: (No Response)

Reviewer #3: No work needed

**Part III – Minor Issues: Editorial and Data Presentation Modifications**

Reviewer #1: (No Response)

Reviewer #3: No work needed

PLOS authors have the option to publish the peer review history of their article (what does this mean?). If published, this will include your full peer review and any attached files.

Reviewer #1: No

Reviewer #3: No

---

## [Editor Report · Acceptance letter]

12 Jul 2023

Dear Dr. Horn,

We are delighted to inform you that your manuscript, "Competition among variants is predictable and contributes to the antigenic variation dynamics of African trypanosomes," has been formally accepted for publication in PLOS Pathogens.

Best regards,

Kasturi Haldar

Editor-in-Chief

PLOS Pathogens

orcid.org/0000-0001-5065-158X

Michael Malim

Editor-in-Chief

PLOS Pathogens

orcid.org/0000-0002-7699-2064